# Untargeted analysis of the serum metabolome in cats with exocrine pancreatic insufficiency

**Patrick C. Barko** [ORCID] *, **David A. Williams**

Department of Veterinary Clinical Medicine, University of Illinois at Urbana-Champaign, Urbana, Illinois, United States of America

* pcbarko@illinois.edu

## Abstract

Exocrine pancreatic insufficiency (EPI) causes chronic digestive dysfunction in cats, but its pathogenesis and pathophysiology are poorly understood. Untargeted metabolomics is a promising analytic methodology that can reveal novel metabolic features and biomarkers of clinical disease syndromes. The purpose of this preliminary study was to use untargeted analysis of the serum metabolome to discover novel aspects of the pathobiology of EPI in cats. Serum samples were collected from 5 cats with EPI and 8 healthy controls. The diagnosis of EPI was confirmed by measurement of subnormal serum feline trypsin-like immunoreactivity (fTLI). Untargeted quantification of serum metabolite utilized ultra-high-performance liquid chromatography-tandem mass spectroscopy. Cats with EPI had significantly increased serum quantities of long-chain fatty acids, polyunsaturated fatty acids, mevalonate pathway intermediates, and endocannabinoids compared with healthy controls. Diacylglycerols, phosphatidylethanolamines, amino acid derivatives, and microbial metabolites were significantly decreased in cats with EPI compared to healthy controls. Diacyclglycerols and amino acid metabolites were positively correlated, and sphingolipids and long-chain fatty acids were negatively correlated with serum fTLI, respectively. These results suggest that EPI in cats is associated with increased lipolysis of peripheral adipose stores, dysfunction of the mevalonate pathway, and altered amino acid metabolism. Differences in microbial metabolites indicate that feline EPI is also associated with enteric microbial dysbiosis. Targeted studies of the metabolome of cats with EPI are warranted to further elucidate the mechanisms of these metabolic derangements and their influence on the pathogenesis and pathophysiology of EPI in cats.

## Introduction

Exocrine pancreatic insufficiency (EPI) is a maldigestive and malabsorptive syndrome caused by insufficient secretion of pancreatic digestive enzymes from pancreatic acinar cells [1]. Retrospective studies have revealed that the most common clinical sign of EPI is weight loss, which affects as many as 91% of patients, but cats with EPI also present with a variety of other clinical signs including diarrhea, anorexia, polyphagia, and unkempt hair-coat [1, 2]. Feline EPI is diagnosed

**Data Availability Statement:** The data underlying the results presented in the study are available from our Mendeley data repository: https://data.mendeley.com/datasets/jnmjt3rnpz/1. The R code underlying the results presented in the study are

available from our GitHub repository: https://github.com/pcbarko/Barko-fEPI-Metabolome.

**Funding:** The authors received no specific funding for this work.

**Competing interests:** The authors have declared that no competing interests exist.

by measuring serum feline trypsin-like immunoreactivity (fTLI) which reflects the concentration of circulating trypsinogen, a protease manufactured and stored only in pancreatic acinar cells. fTLI is a highly sensitive and specific marker for exocrine pancreatic mass and a value < 8 μg/L is diagnostic for EPI [3, 4]. EPI was traditionally thought to be rare in cats, but rates of diagnosis have increased since the introduction and validation of the fTLI assay in 1995 [5].

Feline EPI is a poorly studied phenomenon and its etiopathogenesis is unknown. Feline EPI is commonly thought to be the end-stage chronic pancreatitis, however there is scant clinical or histopathologic evidence in support of this hypothesis [6]. Pancreatic acinar atrophy (PAA), the most common cause of EPI in dogs, has been described in association with EPI in a small number of cats [2]. In addition to those reported in the peer-reviewed literature, the authors have observed several additional cases in which EPI was clearly associated with PAA in the absence of any other histologic evidence of pancreatitis or pancreatic fibrosis. Whether PAA observed in these cases was the end-stage of chronic pancreatitis was not determined, though the conspicuous absence of pancreatic fibrosis suggests an alternative pathogenesis. A lack of detailed histopathologic studies of the pancreata of cats with EPI leaves many unanswered questions regarding its pathogenesis. Previous studies in humans and other mammals have found that a variety of metabolic and nutritional insults can induce PAA. In previous studies protein-calorie malnutrition and deficiencies in copper, zinc, selenium, choline have been associated with pancreatic acinar atrophy in humans and lab animals [7–14]. Studies in the 1970's and early 1980s identified PAA and pancreatic fibrosis in rats fed liquid elemental diets [15]. Interestingly, the PAA was rare in germ-free rats, but common in those grown in conventional. These findings suggest that exocrine pancreatic injury has a complex pathogenesis involving nutritional, metabolic, and microbiologic factors. Additionally, EPI is the cause of numerous nutritional deficiencies including lipid soluble vitamins and cobalamin in cats and other mammals [1, 16, 17]. Thus, investigations of metabolic, nutritional, and microbial features involved in the pathophysiology of EPI in cats are warranted.

To identify metabolic anomalies in cats with EPI, we utilized untargeted, high-throughput analysis of the serum metabolome. The metabolome is the sum of all small molecules (< 1500 kDa) present in an organism or biologic sample that are precursors, intermediates, and end-products of metabolic processes. Metabolites of endogenous and exogenous (diet, environment, and microbial) origin contribute to the serum metabolome of mammals [18, 19]. Quantification of metabolites in a biologic sample can yield valuable insights into the metabolic phenotypes of disease states by elucidating pathophysiologic mechanisms of disease and facilitating the discovery novel biomarkers.

There is a paucity of clinical and experimental data related to the phenomenon of EPI in cats and very little is known about its pathobiology. The purpose of this preliminary study was to characterize the serum metabolomes of cats with EPI and compare them to healthy cats. We intended to identify metabolites and metabolic pathways with plausible roles in the pathogenesis and pathophysiology of EPI in cats. The authors hypothesized that cats with EPI would have systemic metabolic disturbances that would be detectable via analysis of the serum metabolome. Furthermore, we hypothesized that metabolites and metabolic pathways associated with serum fTLI will be candidate biomarkers for the pathogenesis and pathophysiology of the disorder.

## Materials and methods

### Ethics statement

All sample collection methods were approved by the University of Illinois at Urbana-Champaign Institutional Animal Care and Use Committee (UIUC IACUC; Protocol IDs 16147 and 14256). All participating cat owners provided written consent prior to sample collection.

## Animals and sample collection methods

Client-owned cats with a historical diagnosis of EPI were identified from an online forum (epiincats.webs.com) and from primary care veterinary clinics in Champaign County, IL. The medical records of these cats were reviewed by the investigators. Cats were eligible for inclusion if they had a historical serum fTLI concentration ≤8 μg/L, were receiving pancreatic enzyme supplementation, and had historic clinical signs of EPI including weight loss, anorexia, diarrhea, or some combination thereof. These details were confirmed by a review of each cat's medical record, however the investigators were unable to perform physical examinations due to the samples being collected off-site. Fasted serum samples were collected by primary care veterinarians via jugular venipuncture, frozen on site immediately after collection, and shipped on dry ice to the investigators. The samples were stored at -80˚C prior to analysis. The diagnosis of EPI was confirmed in these cats by measuring serum fTLI concentrations <8 μg/L. Serum fTLI radioimmunoassays were performed at the Texas A&M Gastrointestinal Laboratory (College Station, TX). Serum samples from 8 healthy cats from a research colony were used as a control group. Fasted serum samples from these cats were collected by jugular venipuncture, immediately frozen, and banked at -80˚C prior to analysis. Serum samples from these cats were collected as a part of an unrelated study. These samples had been stored for 12 months at -80˚C prior to utilization in the present study.

## Untargeted analysis of serum metabolomes

Serum metabolite profiles were generated via ultrahigh performance liquid chromatography-tandem mass spectroscopy (UPLC-MS/MS) by a commercial laboratory (Metabolon Inc., Morrisville, NC). Serum samples were deproteinated using methanol precipitation with vigorous shaking for 2 min followed by centrifugation. The resulting extract was separated into fractions: two for reverse phase (RP)/UPLC-MS/MS with positive ion mode electrospray ionization (ESI), one for analysis by RP/UPLC-MS/MS with negative ion mode ESI, one for analysis by HILIC/UPLC-MS/MS with negative ion mode ESI. Analytic controls consisting of pooled samples, generated from small volumes of each experimental sample, were analyzed simultaneously with experimental samples to serve as a technical replicate throughout the data set. Methanol-extracted water samples served as process blanks and quality-control standards that were known to not interfere with the measurement of endogenous compounds were spiked into every analyzed sample, allowing monitoring of instrument performance, and aiding chromatographic alignment. Experimental samples were randomized across the platform run with quality control samples spaced evenly among the injections. For UPLC-MS/MS, all methods utilized a Waters ACQUITY ultra-performance liquid chromatographer and a Thermo Scientific Q-Exactive high resolution/accurate mass spectrometer interfaced with a heated electrospray ionization source and Orbitrap mass analyzer operated at 35,000 mass resolution. The sample extracts were dried and reconstituted in solvents compatible to each of the four methods UPLC-MS/MS methods. Each solvent contained standards at fixed concentrations to ensure injection and chromatographic consistency. One aliquot was analyzed using acidic positive ion conditions optimized for hydrophilic compounds. In this method, the extract was gradient eluted from a C18 column (Waters UPLC BEH C18-2.1x100 mm, 1.7 μm) using water and methanol, containing 0.05% perfluoropentanoic acid (PFPA) and 0.1% formic acid (FA). Another aliquot was analyzed using acidic positive ion conditions chromatographically optimized for hydrophobic compounds. In this method, the extract was gradient eluted from the same afore mentioned C18 column using methanol, acetonitrile, water, 0.05% PFPA and 0.01% FA. Another aliquot was analyzed using basic negative ion optimized conditions using a separate dedicated C18 column. The basic extracts were gradient eluted from the

column using methanol and water with 6.5mM Ammonium Bicarbonate at pH 8. The fourth aliquot was analyzed via negative ionization following elution from a HILIC column (Waters UPLC BEH Amide 2.1x150 mm, 1.7 μm) using a gradient consisting of water and acetonitrile with 10mM Ammonium Formate, pH 10.8. The MS analysis alternated between MS and data-dependent MSn scans using dynamic exclusion. The scan range varied slighted between methods but covered 70–1000 m/z. Data extraction, compound identification, and data processing were performed using a proprietary software platform by Metabolon Inc. Compounds were identified by comparison to library entries of purified and authenticated standards. Metabolites were quantified by measuring the area-under-the-curve of the chromatographic peak. Metabolite abundance data were normalized by median scaling and missing values were imputed with the sample set minimum. Studies conducted by Metabolon, Inc. indicate that, using their global UPLC-MS/MS and proprietary data processing platform, minimum value imputation outperforms alternative methods (unpublished, S1 File).

## Statistical analysis

All statistics were performed in the R language for statistical computing (version 3.6.1) [20]. The serum metabolite data set is archived in our Mendeley Data repository (https://data. mendeley.com/datasets/jnmjt3rnpz/1) and R code used in these analyses are available at our GitHub repository (https://github.com/pcbarko/Barko_fEPI_Metabolome).

Log base 10 transformation was applied to prior to ordination, visualization, and statistical procedures. For unsupervised analysis, a statistical heatmap of metabolite abundance and principal component analysis (PCA) were used. To detect significant differences in individual metabolites between groups, a Welch's two-sample t-test was performed for all metabolites. To detect differentially regulated metabolic pathways between cats with EPI and healthy controls, metabolite set enrichment analysis (MSEA) was performed. For MSEA, we attempted to determine whether pre-defined sets (i.e. sub-pathways) of metabolites showed statistically significant differences between the two clinical phenotypes (EPI vs healthy) of interest. First, each metabolite was ranked by the product of the log2 fold-change and the -log10-transformed p-values derived from the Welch's t-test. In this ranked vector of metabolites, the top end of the list contains metabolites that were significantly up-regulated, and the bottom of the list contains metabolites that were significantly down-regulated. MSEA was then performed using the using the fgsea R package [21]. To identify important drivers of sub-pathway enrichment, we selected the leading-edge metabolites from sub-pathways with a statistically significant normalized enrichment score (NES). These leading-edge metabolites are those that contribute most to the NES and are thus important drivers of differing metabolic phenotypes between groups. To detect serum metabolites that were correlated with pancreatic function we performed Pearson correlation tests between the abundance of each metabolite and serum fTLI, an indirect biomarker of pancreatic function.

To account for multiple statistical comparisons, an estimate of the false discovery rate (FDR; q-value) was calculated for all independent statistical tests using the Benjamini-Hochburg method as previously described [22]. For differences in metabolite concentrations, MSEA, and correlation analyses to be considered significant, the following criteria was used: P-value $< 0.05$ and FDR $<0.2$. As this was a pilot study, we sought to strike a balance between avoiding false discoveries and missing the discovery of actual differences that could inform future targeted investigations. For this reason, we selected a relatively liberal false discovery threshold of FDR$<0.2$. In addition to these criteria, we further restricted significant results from the Welch's t-test to metabolites are relatively large effect size of +/- 1.5 (absolute value of the log2-fold-change $> 0.585$) between the EPI and healthy control groups. Similarly, we

imposed a relatively conservative correlation threshold, focusing only on correlations where the absolute value of Pearson's r was greater than 0.7.

## Results

Six domestic shorthair cats with a historical diagnosis of EPI were initially recruited. One of these cats was excluded after it was found to have serum fTLI concentration within the normal reference range on re-testing. In the remaining 5 cats the diagnosis was successfully confirmed, and their serum samples were included in the study. In the EPI group, 3/5 were neutered males and 2/5 were spayed females. All EPI cats were treated with oral pancreatic enzyme supplementation and were fed different balanced commercial feline diets. 3/5 cats in the EPI had historic diagnoses of chronic enteropathy, supported in 2 patients by histologic confirmation of enteric inflammatory infiltrates and a third by a positive response to treatment with oral prednisolone. These 3 cats had received prednisone within 30 days of sample collection. Samples from eight healthy, unrelated, neutered male, domestic shorthair cats from a research colony were used as a control group. The specific compositions of their diets were not recorded in the caretaker's records. There was no significant difference in the cat's age between the two groups (Table 1; $P = 0.081$). Serum fTLI was significantly lower (Table 1; $P = 0.0016$) in the EPI group compared to healthy controls.

733 biochemicals were detected in the sera of cats in this study. Examination of a statistical heatmap of metabolite abundance revealed global differences in serum metabolite concentrations and a clustering dendrogram of samples based on their Euclidian distances separated cats with EPI from healthy controls (Fig 1A). PCA separated samples from cats with EPI from healthy controls along the first principal component (Fig 1B), consistent with metabolome-wide differences in the abundance of serum biochemicals between groups.

Of the 733 measured serum metabolites, 197 varied significantly between the EPI and healthy control groups (S2 File). Of these, 41 biochemicals were significantly higher and 156 were significantly lower in the EPI group compared with healthy controls. To understand the likelihood of false discoveries in our results, we analyzed descriptive statistics for the FDR values. For metabolites considered to be significant, the maximum FDR was 0.867 (8.67%) and 140/197 (71%) of metabolites had an FDR<0.05. Biochemicals from the lipid and amino acid super-pathways were the most well-represented metabolic super pathways among significant results (Fig 2A).

Seven metabolic sub-pathways were differentially enriched between cats with EPI and healthy controls (S3 File). The long-chain fatty acid (LCFA), polyunsaturated fatty acid (PUFA), mevalonate, endocannabinoid, and branched-chain fatty acid (BCFA) sub-pathways were increased in cats with EPI (Fig 2B). The phosphatidylethanolamine (PE) and diacylglycerol (DAG) sub-pathways were decreased in cats with EPI compared with healthy controls (Fig 2B). We selected the leading-edge metabolites from MSEA to identify important drivers of sub-pathway enrichment. 10-heptadecenoate, docosapentaenoate, mevalonolactone,

**Table 1. Patient population statistics.**

|  |  | N | Median | Mean | SD | Min | Max | P-Value |
|---|---|---|---|---|---|---|---|---|
| **Age (years)** | Healthy | 8 | 11.0 | 11.0 | 0.0 | 11.0 | 11.0 | 0.081 |
|  | EPI | 5 | 10.4 | 11.0 | 0.89 | 9.0 | 11.0 |  |
| **fTLI (µg/L)** | Healthy | 8 | 32.95 | 39.53 | 18.31 | 18.3 | 67.5 | 0.0016 |
|  | EPI | 5 | 1.70 | 1.68 | 0.67 | 0.8 | 2.5 |  |

N, number of animals; SD, standard deviation.

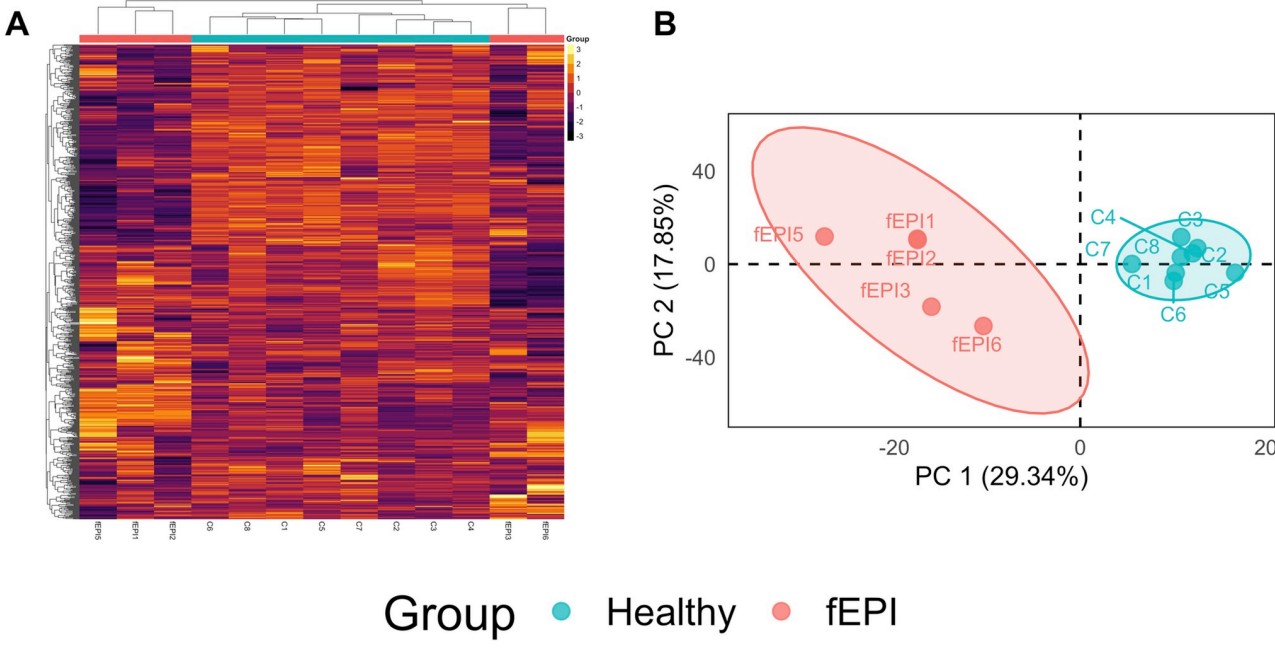

**Fig 1. Unsupervised analysis of serum metabolite profiles. A.** A statistical heatmap of metabolite abundance and clustering dendrogram of samples based on their Euclidian distances revealed separation of cats with EPI from healthy controls. The color of each cell is proportional to the z-scaled abundance of each metabolite. **B.** PCA separates samples from cats with EPI from healthy controls along the first principal component. The colored ellipse surrounding each cluster of samples represents the 95% confidence interval.

N-linolenolyltaurine, and 15-methylpalmitate were the leading-edge metabolites in the LCFA, PUFA, mevalonate, endocannabinoid, and BCFA sub-pathways, respectively (Fig 3A–3E). Palmitoyl-oleoyl-glycerol and 1,2-dilinoleoyl-GPE were drivers of DAG and PE sub-pathway pathway enrichment, respectively (Fig 3F and 3G).

We identified several other important differences in serum metabolite concentrations among groups, predominantly within the amino acid and lipids super pathways (S4 File). Several amino acids including lysine, glycine, proline, and arginine were significantly lower in the sera of cats with EPI compared with healthy controls. N-acetyl-L-amino acid metabolites (N-acetylglutamate, N-acetylserine, N-acetyllysine) and gamma-glutamyl amino acids (were significantly lower in the fEPI group compared with controls. The abundance of cystine was significantly higher in the sera cats with EPI compared with healthy controls. Urea cycle intermediates, including ornithine and arginosuccinate, were significantly lower in cats with EPI. Several microbial metabolites were significantly lower in cats with EPI (Fig 4), including benzoate derivatives (2-hydroxyhippurate, 3-hydroxyhippurate, ethylphenyl sulfate), indolpropionate (a tryptophan derivative), and secondary bile acids (taurolithocholate, taurolithocholate 3-sulfate, ursodeoxycholate, taurodeoxycholate).

To identify metabolites that were correlated with exocrine pancreatic function, we calculated the Pearson correlation coefficient (r) between serum cTLI and each serum metabolite. The serum concentrations of 79 metabolites were significantly ($P < 0.05$; FDR < 0.2) and strongly (Pearson's $r > 0.7$) correlated with serum fTLI (S5 File). Of these, 67 metabolites were positively correlated and 12 were negatively correlated with serum fTLI. Lipid metabolites had the most numerous significant correlations with serum fTLI (32/79) followed by amino acids (24/79), xenobiotics (8/79), peptides (6/79), carbohydrates (4/79), cofactors and vitamins (3/79), nucleotides (1/79) and energy (1/79). Among lipid metabolites, DAGs, PEs, were

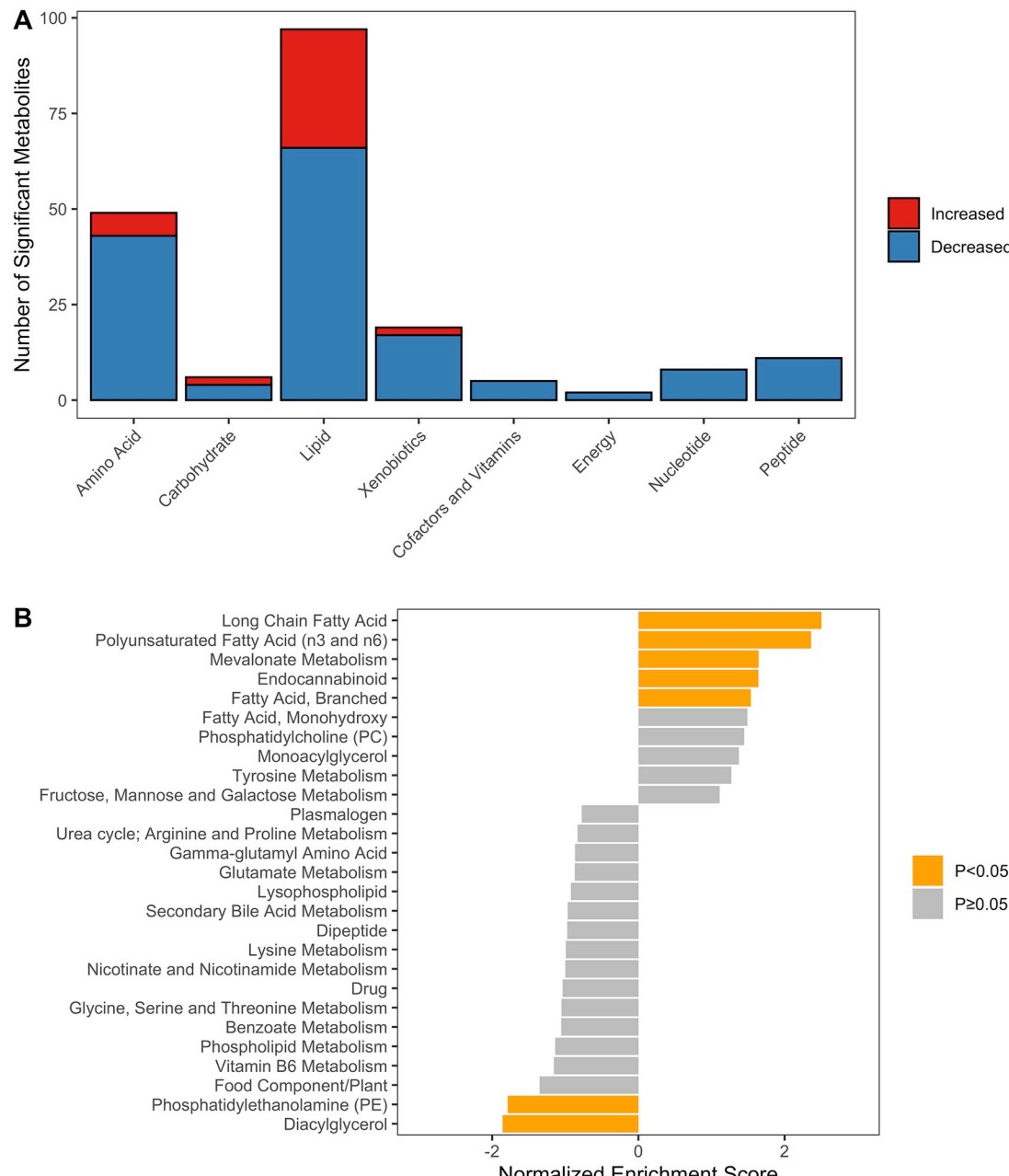

**Fig 2. Metabolite set enrichment analysis. A.** Representation of metabolic super pathways among metabolites that varied significantly between the EPI and healthy control groups. **B.** Metabolite set enrichment analysis (MSEA) was used to detect significantly enriched metabolic sub-pathways in cats with EPI. The normalized enrichment score (NES) represents the degree to which a metabolic sub-pathway is over-represented at the top or bottom list of ranked metabolites. A positive NES means that a metabolic sub-pathway is up-regulated in EPI and a negative NES means that it is down-regulated. The color of the bar representing NES is orange for significantly altered pathways (P<0.05) and grey for those that were not significantly altered (P≥0.05).

positively correlated with serum fTLI whereas sphingolipids and LCFAs were negatively correlated with fTLI (Table 2). Several amino acids including glycine, lysine, and arginine were positively correlated with serum fTLI (Table 3). Numerous other amino acid metabolites including N-acetylated amino acids, dipeptides, gamma-glutamyl amino acids, and urea cycle intermediates were also significantly and positively correlated with serum fTLI (Table 3).

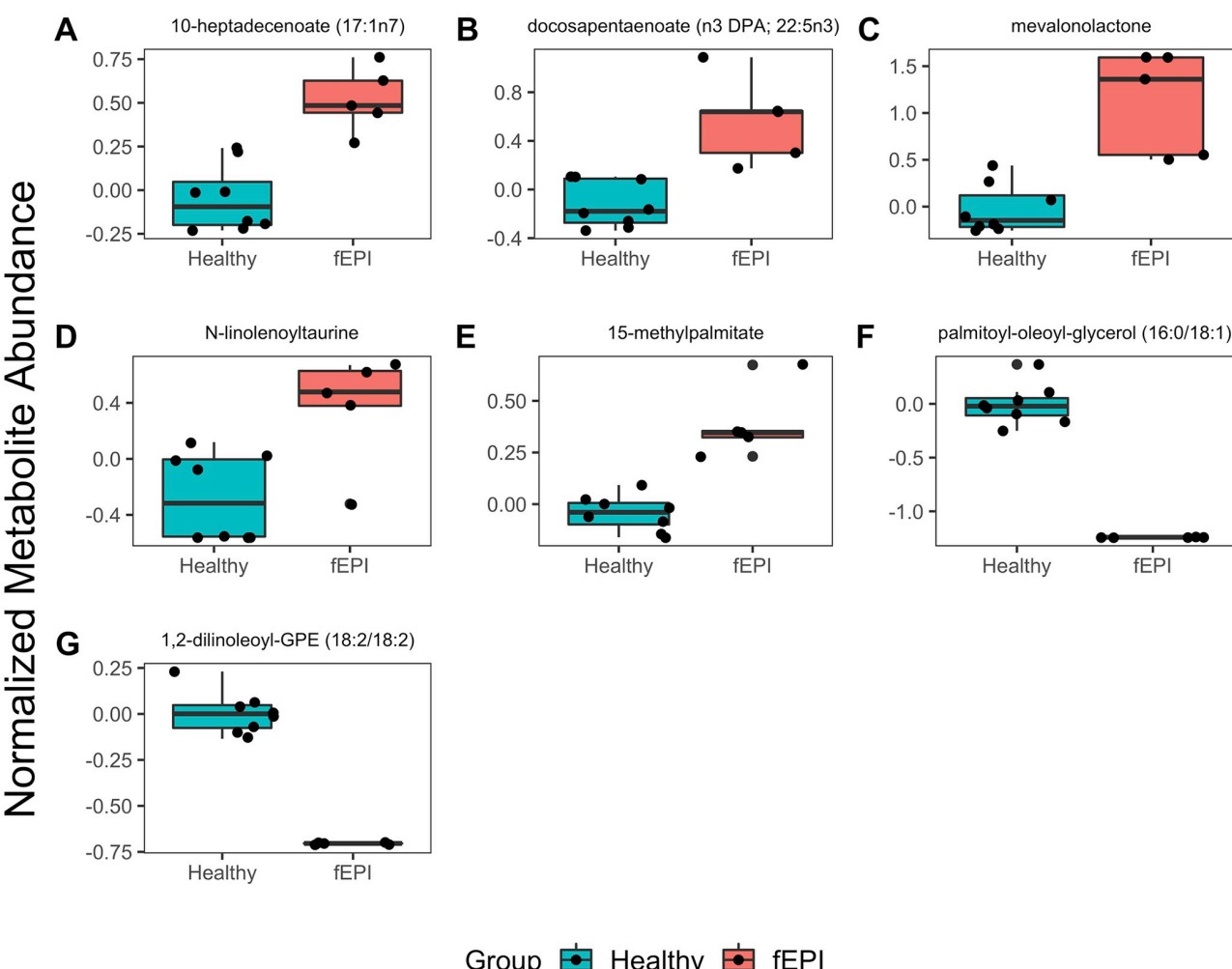

**Fig 3. Comparison of selected metabolites from significantly enriched metabolic sub-pathways.** Selected metabolites were important drivers of metabolic sub-pathway enrichment and varied significantly between cats with EPI and healthy controls. **A.** 10-heptadecenoate (17:1n7), long chain fatty acid; **B.** docosapentanoate (DPA, 22:5n3), polyunsaturated fatty acid; **C.** mevalonolactone, mevalonate metabolism; **D**. N-linolenoyltaurine, endocannabinoid; **E**. 15-methylpalmitate, branched fatty acid; **F.** palmitoyl-oleoyl-glycerol (16:0/18:1), diacyclglycerol; **G.** 1, 2-dilinoleoyl-GPE (18:2/18:2), phophatidylethanolamine.

## Discussion

This study identified significant differences in numerous serum metabolites and metabolic pathways between cats with EPI and healthy controls. Overall, 26.9% (197/733) of all serum metabolites varied significantly between groups and PCA separated cats with EPI from healthy controls. These findings indicate that there are substantial differences in serum metabolites between cats with EPI and healthy controls. However, PCA also revealed that cats with EPI appeared to have higher inter-individual variation than healthy controls. Whether this is a feature of variability within the metabolic phenotype of EPI in cats or a consequence of genetic or environmental factors (diet, housing, comorbidities, etc.) is unknown. Nonetheless, multiple biochemical pathways were significantly different in cats with EPI compared with healthy controls. In this small cohort of cats, the metabolic phenotype associated with EPI was characterized by increased LCFAs, PUFAs, endocannabinoids, mevalonate pathway intermediates and branched-chain fatty acids; and decreased DAG, PE, amino acids, urea cycle intermediates,

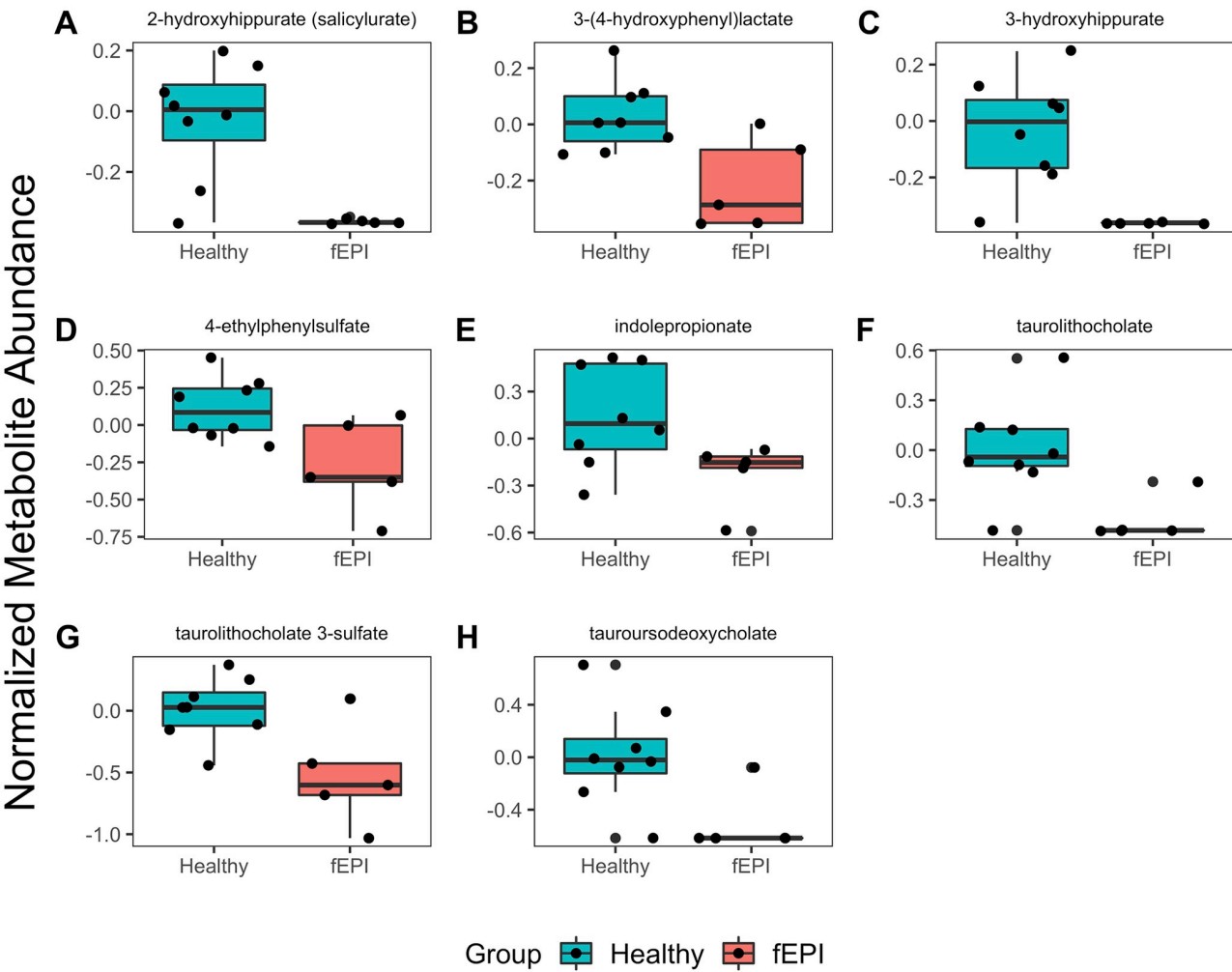

**Fig 4. Comparison of serum microbial metabolites between cats with EPI and healthy controls.** The abundance of several serum microbial metabolites were decreased in cats with EPI compared with healthy controls. On the y-axis are the normalized (median scaled and log-transformed) abundance of each serum metabolite.

and microbial metabolites. These findings demonstrate that analysis of the metabolome may be a useful investigative approach to describe pathophysiologic aspects of EPI in domestic cats.

Of all the metabolic pathways associated with EPI, disturbances in lipid metabolism were the most profound. LCFAs, PUFAs, and branched-chain fatty acids were significantly increased, and DAGs were significantly decreased in cats with EPI (S4 File). As dietary fat malabsorption is a typical feature of EPI, the presence of altered lipid metabolism is not surprising. Fasted serum free fatty acid concentrations reflect the degree of lipolysis of peripheral adipose stores [23]. Lipolysis is a tightly regulated process stimulated by hormones such as norepinephrine, cortisol, and glucagon, and inhibited by insulin [24, 25]. In a fasted state, and during periods of physiologic stress including malnutrition, hydrolysis of trigylcerides in adipose tissue is the primary source of circulating LCFAs and PUFAs [23, 25]. Metabolites associated with beta-oxidation of FFAs, myristoylcarnitine, 3-hydroxydecanoate, and 3-hydroxylaurate, were also significantly increased in the serum of cats with EPI [26, 27]. Based on these observations, increased FFA and PUFA levels in cats with EPI are likely due to lipolysis of triglycerides from peripheral adipose tissue reserves to maintain energy homeostasis. This interpretation is

**Table 2. Significant correlations among lipid metabolites and serum fTLI.**

| Biochemical Name | Sub-Pathway | Pearson's r | P-Value | FDR |
|---|---|---|---|---|
| palmitoyl-oleoyl-glycerol (16:0/18:1) | Diacylglycerol | 0.82 | 0.000554 | 0.01238949 |
| oleoyl-linoleoyl-glycerol (18:1/18:2) | Diacylglycerol | 0.81 | 0.000792 | 0.01238949 |
| oleoyl-oleoyl-glycerol (18:1/18:1) | Diacylglycerol | 0.81 | 0.000726 | 0.01238949 |
| palmitoyl-linoleoyl-glycerol (16:0/18:2) | Diacylglycerol | 0.81 | 0.000704 | 0.01238949 |
| diacylglycerol (16:1/18:2, 16:0/18:3) | Diacylglycerol | 0.8 | 0.000983 | 0.01300897 |
| stearoyl-linoleoyl-glycerol (18:0/18:2) | Diacylglycerol | 0.79 | 0.00116 | 0.01411373 |
| oleoyl-linoleoyl-glycerol (18:1/18:2) | Diacylglycerol | 0.78 | 0.00171 | 0.01763576 |
| palmitoyl-arachidonoyl-glycerol (16:0/20:4) | Diacylglycerol | 0.77 | 0.00204 | 0.01763576 |
| palmitoyl-oleoyl-glycerol (16:0/18:1) | Diacylglycerol | 0.76 | 0.00269 | 0.01999622 |
| palmitoyl-linoleoyl-glycerol (16:0/18:2) | Diacylglycerol | 0.75 | 0.00299 | 0.02033039 |
| N-linolenoyltaurine | Endocannabinoid | -0.82 | 0.00066 | 0.01238949 |
| 15-methylpalmitate | Fatty Acid, Branched | -0.73 | 0.00434 | 0.02283922 |
| 3-methyladipate | Fatty Acid, Dicarboxylate | 0.74 | 0.0035 | 0.02129227 |
| glycerophosphoglycerol | Glycerolipid Metabolism | 0.75 | 0.00308 | 0.02033039 |
| myo-inositol | Inositol Metabolism | 0.82 | 0.000553 | 0.01238949 |
| 10-nonadecenoate (19:1n9) | Long Chain Fatty Acid | -0.74 | 0.00382 | 0.0216363 |
| 10-heptadecenoate (17:1n7) | Long Chain Fatty Acid | -0.82 | 0.000659 | 0.01238949 |
| 1,2-dilinoleoyl-GPE (18:2/18:2) | Phosphatidylethanolamine | 0.84 | 0.000288 | 0.01238949 |
| 1-palmitoyl-2-linoleoyl-GPE (16:0/18:2) | Phosphatidylethanolamine | 0.75 | 0.00289 | 0.02019983 |
| 1-stearoyl-2-linoleoyl-GPE (18:0/18:2) | Phosphatidylethanolamine | 0.74 | 0.0042 | 0.02283922 |
| 1,2-dioleoyl-GPE (18:1/18:1) | Phosphatidylethanolamine | 0.73 | 0.00465 | 0.02314497 |
| 1-linoleoyl-2-arachidonoyl-GPE (18:2/20:4) | Phosphatidylethanolamine | 0.73 | 0.00424 | 0.02283922 |
| 1-palmitoyl-2-arachidonoyl-GPE (16:0/20:4) | Phosphatidylethanolamine | 0.72 | 0.00508 | 0.02350461 |
| 1-palmitoyl-2-oleoyl-GPE (16:0/18:1) | Phosphatidylethanolamine | 0.72 | 0.00576 | 0.02489755 |
| docosapentaenoate (n3 DPA; 22:5n3) | Polyunsaturated Fatty Acid | -0.76 | 0.00248 | 0.01999622 |
| sphinganine | Sphingolipid Metabolism | 0.74 | 0.00362 | 0.02162188 |
| sphingomyelin (d18:2/23:1) | Sphingolipid Metabolism | -0.76 | 0.00275 | 0.01999622 |
| sphingomyelin (d18:2/14:0, d18:1/14:1) | Sphingolipid Metabolism | -0.77 | 0.0019 | 0.01763576 |
| sphingomyelin (d18:0/18:0, d19:0/17:0) | Sphingolipid Metabolism | -0.78 | 0.0018 | 0.01763576 |
| sphingomyelin (d18:1/14:0, d16:1/16:0) | Sphingolipid Metabolism | -0.82 | 0.000602 | 0.01238949 |
| myristoyl dihydrosphingomyelin (d18:0/14:0) | Sphingolipid Metabolism | -0.91 | 1.75E-05 | 0.00574891 |

FDR, false discovery rate.

consistent with the clinical picture of feline EPI, which is characterized by cachexia and decreased body condition due to loss of peripheral adipose tissue.

Cats with EPI were found to have significantly increased serum mevalonate and mevalono-lactone compared with healthy cats. Mevalonate is product of HMG-CoA reductase and the first metabolic intermediate in the mevalonate pathway, through which vital biochemicals, including cholesterol, are formed [28]. While disorders in the mevalonate pathway have not been described in cats, mevalonate kinase deficiency (MKD), an inborn error of metabolism caused by decreased mevalonate kinase (MK) activity, causes mevalonate accumulation in humans is [29]. MKD is associated with chronic inflammatory conditions such as recurrent fever, polyarthritis, and sterile peritonitis [30, 31]. Immune dysfunction in patients with MKD is thought to be induced by increased secretion of pro-inflammatory cytokines (IL-1β, TNFα) and mitochondrial dysfunction.[32, 33] Chronic pancreatitis, considered a common cause of EPI in cats, is often associated with other inflammatory conditions including inflammatory

**Table 3. Significant correlations among amino acid metabolites and fTLI in the serum.**

| Biochemical Name | Sub-Pathway | Pearson's r | P-Value | FDR |
|---|---|---|---|---|
| N-acetylalanine | Alanine and Aspartate | 0.73 | 0.00438 | 0.02283922 |
| 2-hydroxybutyrate | Glutathione | 0.73 | 0.00505 | 0.02350461 |
| glycine | Glycine, Serine and Threonine Metabolism | 0.77 | 0.00198 | 0.01763576 |
| N-acetylserine | Glycine, Serine and Threonine Metabolism | 0.77 | 0.0023 | 0.01937363 |
| N-acetylglycine | Glycine, Serine and Threonine Metabolism | 0.73 | 0.00457 | 0.02309673 |
| 1-methylguanidine | Guanidino and Acetamido Metabolism | 0.73 | 0.00478 | 0.02343693 |
| histamine | Histidine Metabolism | 0.74 | 0.00381 | 0.0216363 |
| 3-methylhistidine | Histidine Metabolism | 0.71 | 0.007 | 0.02910842 |
| isobutyrylglycine | Leucine, Isoleucine and Valine Metabolism | 0.72 | 0.00506 | 0.02350461 |
| pipecolate | Lysine Metabolism | 0.85 | 0.000225 | 0.01238949 |
| 2-aminoadipate | Lysine Metabolism | 0.81 | 0.000735 | 0.01238949 |
| 6-oxopiperidine-2-carboxylate | Lysine Metabolism | 0.78 | 0.00173 | 0.01763576 |
| N6-acetyllysine | Lysine Metabolism | 0.77 | 0.00196 | 0.01763576 |
| 5-hydroxylysine | Lysine Metabolism | 0.75 | 0.00328 | 0.02033039 |
| lysine | Lysine Metabolism | 0.75 | 0.00326 | 0.02033039 |
| N6,N6,N6-trimethyllysine | Lysine Metabolism | 0.75 | 0.00312 | 0.02033039 |
| methionine sulfone | Methionine, Cysteine, SAM and Taurine Metabolism | 0.73 | 0.00495 | 0.02350461 |
| phenylpyruvate | Phenylalanine Metabolism | 0.77 | 0.00198 | 0.01763576 |
| N-delta-acetylornithine | Urea cycle; Arginine and Proline Metabolism | 0.85 | 0.000206 | 0.01238949 |
| N-monomethylarginine | Urea cycle; Arginine and Proline Metabolism | 0.8 | 0.00104 | 0.01314037 |
| N-acetylarginine | Urea cycle; Arginine and Proline Metabolism | 0.76 | 0.00267 | 0.01999622 |
| dimethylarginine | Urea cycle; Arginine and Proline Metabolism | 0.75 | 0.00327 | 0.02033039 |
| arginine | Urea cycle; Arginine and Proline Metabolism | 0.72 | 0.0055 | 0.02441623 |
| 2-oxoarginine | Urea cycle; Arginine and Proline Metabolism | 0.71 | 0.00664 | 0.02796541 |
| 4-hydroxyphenylacetylglycine | Acetylated Peptides | 0.81 | 0.000864 | 0.01274045 |
| prolylglycine | Dipeptide | 0.76 | 0.00264 | 0.01999622 |
| leucylglycine | Dipeptide | 0.71 | 0.00647 | 0.02760331 |
| N-acetylcarnosine | Dipeptide Derivative | 0.81 | 0.000756 | 0.01238949 |
| anserine | Dipeptide Derivative | 0.72 | 0.00541 | 0.02434569 |
| gamma-glutamylglycine | Gamma-glutamyl Amino Acid | 0.83 | 0.000431 | 0.01238949 |

FDR, false discovery rate.

bowel disease, cholangitis, and interstitial nephritis [34]. Given the association between mevalonate accumulation and multifocal inflammatory disorders in humans, dysfunction of the mevalonate pathway could play a role in the pathogenesis of chronic pancreatitis and EPI in cats. Interestingly, a recent study of the serum metabolome of dogs with EPI conducted by our group did not detect any differences in biochemicals in the mevalonate pathway between dogs with EPI and healthy controls (S1 Fig). This raises the possibility that dysfunction of the mevalonate pathway may be a unique feature of the pathobiology of EPI in cats, and not a sequala of EPI generally. Targeted investigations of the mevalonate pathway in a larger group of cats with EPI, pancreatitis, inflammatory bowel disease, and cholangitis are currently underway.

With few exceptions, amino acid metabolites and urea cycle intermediates that differed between groups were lower in the serum of cats with EPI (S4 File). Numerous amino acids and urea cycle intermediates were also positively correlated with serum fTLI, a biomarker for pancreatic acinar mass. Ammonia generated by amino acid catabolism is converted to urea via the urea/ornithine cycle in the liver [35]. These findings are indicative of decreased flux through

the urea cycle, possibly a compensatory response to limited amino acid availability for catabolic processes. This interpretation is supported by the observation of decreased serum N-acetylglutamate, an allosteric activator of the urea cycle, in cats with EPI [36]. Decreased serum amino acids in cats with EPI could reflect their reduced availability due to dietary protein malabsorption caused by deficient secretion of pancreatic proteases, malabsorption caused by enteric dysfunction, decreased muscle mass due to cachexia, or some combination thereof. As EPI is known to reduce dietary protein digestion and absorption, it is likely that this contributed to our findings [4]. Similarly, weight loss, including loss of lean muscle mass, is a characteristic clinical sign of EPI in cats and a likely contributor to defects in amino acid catabolism. It is also possible that an amino acid deficiency could contribute to the pathogenesis of EPI. Previous studies have shown that protein-calorie malnutrition induces pancreatic dysfunction and PAA, an affect that is likely mediated by a reduction in amino acid availability [7–10]. Interestingly, EPI is commonly identified in humans with celiac and IBD, both of which may cause defects in dietary amino acid absorption and limit the rate of synthesis of proteins by pancreatic acini [9, 11, 12]. Chronic enteropathies that affect enteric mucosal absorptive function are relatively common in cats with EPI, as evinced by the observation of low serum folate in many cats with EPI. Of the five cats with EPI sampled for this study, two had histologically confirmed diagnoses of inflammatory bowel disease and a third had a clinical history consistent with a chronic enteropathy. It is plausible that, as in humans, chronic enteric dysfunction could contribute to exocrine pancreatic dysfunction in cats by altering amino acid availability.

We also observed that γ-glutamyl amino acids were significantly decreased in cats with EPI. Gamma-glutamyltransferase (GGT) is a membrane-bound enzyme that catalyzes the transfer of a γ-glutamyl moiety from glutathione conjugates to amino acid carriers, generating γ-glutamyl amino acids. Gamma-glutamyl amino acids are transported across the cell membrane and used for intracellular generation of cysteine and de-novo synthesis of glutathione, leaving free amino acids as by-products [37]. Studies in rats and other mammals have shown that GGT is expressed abundantly on the apical membrane of pancreatic acinar cells and in pancreatic ductal cells [38, 39]. With respect to protein synthesis, the exocrine pancreas is among the most active tissues in the mammalian body, requiring a high rate of amino acid transport to support the synthesis and secretion of digestive enzymes. Previous studies have found that the γ-glutamyl cycle maintains intracellular concentrations of amino acids and glutathione in the exocrine pancreas [40–42]. In the present study, decreased serum concentrations of γ-glutamyl cycle byproducts including γ-glutamyl amino acids, 5-oxoproline, and cysteineylglycine were evidence of dysfunction in the γ-glutamyl cycle in cats with EPI. We also identified evidence of oxidative stress in cats with EPI including significantly increased cystine, in addition to oxidized glutathione and cysteine-glutathione disulfide, both of which were increased in cats with EPI but were not statistically significant. These findings could be explained by reduced activity of GGT. Furthermore, it is plausible that dysfunction in the γ-glutamyl cycle could play a role in the pathogenesis of EPI by altering intracellular amino acid availability for digestive enzyme synthesis, increasing susceptibility to oxidative damage, or both.

We also observed significantly decreased concentrations of microbial metabolites in the serum of cats with EPI. Small intestinal dysbiosis is associated with EPI in dogs [43, 44]. Though no studies of the enteric microbiome have been conducted in cats with EPI, it is assumed to be a feature of the disorder due to malabsorption of dietary macromolecules and the presence of diarrhea in many affected cats. Indeed, this study identified significant differences in several microbial metabolites in the serum of cats with EPI compared with healthy cats. Taurolithocholate 3-sulfate, a secondary bile acid (BA) was decreased in cats with EPI. Secondary BAs are generated via the bacterial 7-dehydroxylation of primary bile acids, a reaction that only occurs in some species of *Clostridium* and *Eubacteria* [45–47]. Recent studies

have found that fecal secondary BAs, as well as the relative abundance *Clostridium* and *Eubacteria* are reduced in dogs with EPI compared to healthy controls [43, 44]. In addition to secondary BAs, microbial metabolites of tryptophan, lysine, benzoate, and histidine were significantly different between cats with EPI and healthy controls. These biochemical have been identified and linked to microbial metabolism in recent metabolomics investigations [18, 48, 49]. Collectively, these results indicate that enteric dysbiosis is a feature of feline EPI and that microbial metabolites contribute significantly to serum metabolite profiles in cats.

These findings should be interpreted considering several limitations. First, the number of animals in both groups was low. Over a period of 12 months, we were able to identify 6 cats with a historic diagnosis of EPI and confirm the diagnosis in 5/6 cats. Additionally, this was an unfunded pilot study and our limited resources limited us to analyzing a maximum of 6 samples from cats with EPI, in addition to the 8 healthy controls. In the author's opinion, EPI is under-diagnosed because its clinical signs mimic more common digestive diseases, most notably chronic enteropathies. As clinical signs of gastrointestinal dysfunction are not specific to an underlying etiology, all cats with clinical signs of gastrointestinal dysfunction should be tested for EPI via assay of serum fTLI. The true prevalence of EPI in cats is unknown and expanded testing would facilitate the identification of larger numbers of patients for prospective studies. Other limitations are related to environmental differences among the subjects. The healthy cats used as controls were housed in a vivarium rather than a home setting like the cats in the EPI group, and the commercial diets fed to the two groups differed. All cats in the control group were receiving pancreatic enzyme supplements at the time of sample collection, whereas none of the healthy cats were given pancreatic enzymes. This is unlikely to have contributed significantly to our findings as data from our group suggests that enzyme supplementation has a minimal impact on the serum metabolome and does not significantly impact lipid or amino acid metabolism in dogs (S1 Fig). Additionally, all the cats in the control group were male, whereas most cats in the EPI group were female. Studies in humans have revealed a significant effect of sex on serum metabolome profiles in humans and dogs [50, 51]. The effects of sex on specific metabolites has not been studies in cats, however some of the metabolites we detected in association with EPI are affected by sex in humans. Amino acids (especially branched-chain amino acids), urea cycle intermediates, and ɣ-glutamyl amino acids have been found to be higher in males and some fatty acids higher in females [50]. Thus, the authors cannot exclude the possibility that differences in sex could have impacted our findings. Two of the cats in the EPI group received prednisolone intermittently prior to sample collection. A previous study of humans receiving dexamethasone revealed a significant impact of glucocorticoid administration on the serum metabolomes of healthy humans [52]. However, the effects of dexamethasone were opposite to those observed in this study; serum PUFAs decreased and amino acids increased after treatment. For this reason, prednisolone administration is unlikely to have caused the serum metabolite disturbances identified in this study of feline EPI. Finally, it is possible that comorbidities associated with EPI in cats could have affected our results. In the EPI group two cats had historic diagnoses of IBD and one had steroid-responsive chronic enteropathy (CE). Chronic enteropathies are common comorbidities in cats with EPI but their effects on serum metabolomes in cats with EPI are not known. It is possible that some of the differences we observed between the groups could have been driven by the presence of comorbid conditions and/or enteric microbiota dysbiosis associated with IBD/CE. However, the small sample size in our study means it would be impossible to determine whether there are significant differences among cats with EPI alone and those with EPI+IBD that could have affected our results. Despite these limitations, this preliminary study is a valuable contribution to the state of knowledge of EPI in cats. By identifying novel biochemical targets associated with EPI, we have provided new insights for targeted follow-up studies.

## Conclusions

This study revealed differences in several biochemical pathways in cats with EPI. While it is likely that some of these metabolic disturbances are secondary to EPI, it is possible that some may also play a role in the pathogenesis and pathophysiology of EPI. Increased long-chain fatty acids and polyunsaturated fatty acids are indicative of lipolysis of peripheral adipose stores and may be involved in the pathophysiology of weight loss in affected cats. Lower serum levels of amino acids and urea cycle intermediates are consistent with reduced amino acid availability, decreased protein catabolism, or both. Several biochemicals associated with microbial metabolism are altered in cats with EPI which is evidence of an impact microbial metabolism on the metabolome of cats with EPI. Additional studies of the metabolome and microbiome in cats with EPI are warranted to confirm the validity of these findings and elucidate their pathophysiologic significance.

## Supporting information

**S1 File. Comparison of imputation methods for missing values.** Results of internal validation studies conducted by Metabolon Inc.
(DOCX)

**S2 File. Complete results of Welch's test for differences in metabolite abundance.** This comma-separated value file contains the full output of the statistical comparison of metabolite abundance data between cats with EPI and healthy controls.
(CSV)

**S3 File. Complete results of metabolite set enrichment analysis.** This comma-separated value file contains the full output of metabolite set enrichment analysis (MSEA) to detect significantly enriched metabolic sub-pathways between cats with EPI and healthy controls.
(CSV)

**S4 File. Tables of significantly altered lipid and amino acid metabolites.**
(DOCX)

**S5 File. Complete results of correlation analysis.** This comma-separated value file contains the full output of the Spearman rank correlation test to detect serum metabolites that were significantly correlated with serum fTLI.
(CSV)

**S1 Fig. Comparison of serum mevalonolactone concentrations in dogs with EPI vs healthy controls.** The wilxocon rank-sum test was used to compare the abundance of mevalonolactone in the sera of dogs with EPI and healthy dogs (PostEnz). Unlike cats with EPI, there is no significant difference in the abundance of mevalonolactone in the sera of dogs with EPI compared with healthy controls.
(TIF)

## Acknowledgments

The authors wish to acknowledge Carol Pilger, Olesia Kennedy and Drs. Anisha Jambhekar, Catherine Williams, and Debra Hatman, for their assistance with this study.

## Author Contributions

**Conceptualization:** Patrick C. Barko, David A. Williams.

**Data curation:** Patrick C. Barko.

**Formal analysis:** Patrick C. Barko.

**Funding acquisition:** David A. Williams.

**Investigation:** Patrick C. Barko.

**Methodology:** Patrick C. Barko.

**Project administration:** David A. Williams.

**Software:** Patrick C. Barko.

**Visualization:** Patrick C. Barko.

**Writing – original draft:** Patrick C. Barko.

**Writing – review & editing:** Patrick C. Barko, David A. Williams.

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
