## [Decision Letter · Decision Letter 0]

13 Jul 2021

PONE-D-21-17436

Untargeted analysis of the serum metabolome in cats with exocrine pancreatic insufficiency

PLOS ONE

Dear Dr. Barko,

Thank you for submitting your manuscript to PLOS ONE. After careful consideration, we feel that it has merit but does not fully meet PLOS ONE’s publication criteria as it currently stands. Therefore, we invite you to submit a revised version of the manuscript that addresses the points raised during the review process, as stated in the reviews.

Please submit your revised manuscript within 3 months. If you will need more time than this to complete your revisions, please reply to this message or contact the journal office at plosone@plos.org. Please include the following items when submitting your revised manuscript:

We look forward to receiving your revised manuscript.

Kind regards,

Francisco X. Real

Academic Editor

PLOS ONE

Journal Requirements:

1. Please ensure that your manuscript meets PLOS ONE's style requirements, including those for file naming. The PLOS ONE style templates can be found athttps://journals.plos.org/plosone/s/file?id=wjVg/PLOSOne_formatting_sample_main_body.pdf and https://journals.plos.org/plosone/s/file?id=ba62/PLOSOne_formatting_sample_title_authors_affiliations.pdf

Additional Editor Comments (if provided):

Reviewers' comments:

Reviewer's Responses to Questions

**Comments to the Author**

1. Is the manuscript technically sound, and do the data support the conclusions?

Reviewer #1: Yes

Reviewer #2: Yes

2. Has the statistical analysis been performed appropriately and rigorously? 

Reviewer #1: Yes

Reviewer #2: No

3. Have the authors made all data underlying the findings in their manuscript fully available?

Reviewer #1: Yes

Reviewer #2: Yes

4. Is the manuscript presented in an intelligible fashion and written in standard English?

Reviewer #1: Yes

Reviewer #2: Yes

5. Review Comments to the Author

Reviewer #1: In this study Barko et al compared the metabolome profile of cats with diagnosed exocrine pancreatic insufficiency with control animals. Although the data is preliminary and in the current form does not allow us to draw strong conclusions, it is interesting and highlights the complex differences between healthy animals and those with EPI. The major limitations of the study is correctly listed in the discussion by the authors. It is difficult to judge how the differences between the control and the EPI groups (males in control, females in the EPI group, etc.) affect the results of the metabolome analysis. More detailed description of the different groups may improve the scientific merit.

Major comments:

1. Two animals in the EPI group had inflammatory bowel disease, and one more chronic entheropathy. Would that affect the metabolome? How can we say that the observed difference from the contol group is due to the EPI and not to the IBD? Did the metabolome profile of the cats with EPI and EPI+IBD showed any difference?

2. Had the cats any alterations in the laboratory parameters (such as CRP)? These shall be included to the description of the experimental groups.

3. Did the cats in the EPI group show signs of malabsorption or cahexia? Was the weight of the animals in the two groups different?

4. The authors may consider to extend the study and involve more animals with EPI to increase the robustness of the data.

5. The control group was from a research colony. Did they receive any treatment or other manipulation? It would be more accurate to involve control animals that were kept in a home-like setting and receiving similar diet than EPI group.

Minor comments:

In table 1. Population statistics the unit of the age (year or month) shall be highlighted. Also, sex and weight shall be presented in this table.

Reviewer #2: The article describes the identified significant differences in serum metabolites and metabolic pathways in cats with exocrine pancreatic insufficienty (EPI) and healthy controls. I enjoyed reading it, as it is well explained and easy to follow. I really liked that the authors have the data available in Mendeley and the R code used to analyze it is available in github. However, some steps of the statistical analysis need to be revised prior publication. I recommend a major revison.

Specifically:

Untargeted analysis of serum metabolomes. Authors explain that it was done by UPLC with a commercial laboratory. Authors should state which column was used for the analysis. And which extraction method. Is not the same a methal extraction than a biphasic one. And is not the same using a hilic or a csh column.

The selection of method for handling missing values can significantly affect subsequent data analyses and interpretations. Why the authors have used the minimum? In metabolomics, usually is used the half-minimum or kNN, pqn, ...

Figure 1B fEPI group only show 4 red dots corresponding to 4 samples when there are 5 cases.

Did the authors conduct an exploratory PCA wihtout any normalization or scaling to check for possible outliers?

False discovery rate FDR<0.2 is rather big. This means 20% of false discovery rate is allowed. Why the authors didn’t use a lower FDR?

Why if the control group was constructed from a research cats colony, all the selected ones are males? In the EPI group there are also females. Gender is not important to the authors? Specific gender differences are seen in metabolomics, and this can induce a bias in the study.

Table 1: please use the same significant digits. If they are calculated with 2 signigicant digits (e.x. mean 39.53) give 2 significant digits for the SD (ex. 18.306 should be 18.31). Be consistent in all the talbe fo the same variable (all age results with 1 significant digit, all fTLI with 2 significant digits).

Figures are pixelated. Should be uploaded with better quality.

Table 2. Some compounds are repeates with a [1] or [2] at the end...what do this numbers [1] and [2] mean?

Table 2 and 3 are rather long to be in the main manuscript. I recommend to include them in the supplementary material.

6. PLOS authors have the option to publish the peer review history of their article (what does this mean?). If published, this will include your full peer review and any attached files.

Reviewer #1: **Yes: **József Maléth

Reviewer #2: **Yes: **Raquel Cumeras

---

## [Author Response · Author response to Decision Letter 0]

20 Aug 2021

Reviewer #1: In this study Barko et al compared the metabolome profile of cats with diagnosed exocrine pancreatic insufficiency with control animals. Although the data is preliminary and in the current form does not allow us to draw strong conclusions, it is interesting and highlights the complex differences between healthy animals and those with EPI. The major limitations of the study is correctly listed in the discussion by the authors. It is difficult to judge how the differences between the control and the EPI groups (males in control, females in the EPI group, etc.) affect the results of the metabolome analysis. More detailed description of the different groups may improve the scientific merit.

Reviewer #1, thank you for your thoughtful comments on our manuscript. We agree that a more detailed phenotypic characterization of the EPI and healthy control groups and greater sample sizes would improve the scientific merit of the manuscript. However, this was an unfunded pilot study using a combination of residual sample material from another unrelated investigation (healthy cats) and serum that was sent to us by local primary care veterinarians (EPI group). As such, sample types and quantities of samples available for the study were limited. Likewise, we only had access to the medical records for the EPI cats and were not able to perform physical examinations ourselves. We recognize these important limitations and have discussed them in detail in our “Discussion” section of the manuscript, as you noted. In the interest of full transparency, we have added additional detail to the “Materials and Methods” and “Discussion” sections of the manuscript. As there are no published studies of the pathophysiology of EPI in cats and the disease is poorly understood, we believe that the data we have generated deserves to be published. Our goal in attempting to publish this manuscript is to make our findings available for other investigators so they can inform the conception, design, and interpretation of future investigations of EPI in cats. 

Major comments:

1. Two animals in the EPI group had inflammatory bowel disease, and one more chronic entheropathy. Would that affect the metabolome? How can we say that the observed difference from the control group is due to the EPI and not to the IBD? Did the metabolome profile of the cats with EPI and EPI+IBD showed any difference?

We acknowledge that the presence of IBD and/or chronic enteropathy (CE) in the EPI group could have affected our results. IBD and other chronic enteropathies are common comorbidities in cats with EPI [J Vet Intern Med. 2016 Nov-Dec; 30(6): 1790–1797.]. It is possible that some of the differences we observed between the groups could have been driven by factors other than EPI, including comorbid disorders and/or enteric microbiota dysbiosis. We have updated Figure 1 to include text annotation of each individual cat to help the reviewer understand the potential effect of IBD or chronic enteropathy on our results. Examining the supplemental PCA plot, PC1 separates EPI from healthy controls indicating the major source of variation among samples is group membership (EPI vs healthy). However there appear to be 2 clusters of samples from the EPI group that are separated along PC2. One cluster is formed by three cats: two with IBD (fEPI1, fEPI2) and one non-IBD cat (fEPI5). Another cluster is formed by the cat with CE (fEPI3) and fEPI6 (non-CE). However, the small sample size in our study means precludes the ability to detect significant differences between cats with EPI alone and those with IBD/CE that could have affected our results. Future studies should focus on the effect of comorbidities, including IBD and CE on the serum metabolomes of cats with EPI. We have also expanded briefly on this topic in our “Discussion” section in the revised manuscript.

2. Had the cats any alterations in the laboratory parameters (such as CRP)? These shall be included to the description of the experimental groups.

This investigation was an unfunded pilot study and we were limited in terms of both the sample volume available and costs associated with additional phenotypic characterization. We do not have complete clinicopathologic data available for these cats, nor do we have access to additional sample material to run these assays. We agree that the evaluation of additional laboratory parameters would be useful but were unable to include that in our study design. To the authors knowledge, there is no validated assay available for measuring C-reactive protein (CRP) in cats, but we agree that systemic inflammation can influence serum metabolite concentrations. Future studies could incorporate assessment of feline inflammatory markers (e.g. serum amyloid A which has been validated in cats) but given the funding and logistic limitations of our pilot study, we were unable to do so. We have also expanded briefly on this topic in our “Discussion” section. 

3. Did the cats in the EPI group show signs of malabsorption or cahexia? Was the weight of the animals in the two groups different?

Unfortunately, we do not have access to the weights of the healthy control cats as it was not recorded by the investigator that collected the samples for the original study they were used for. Weight loss is the most common clinical sign of EPI in cats and all cats enrolled in the EPI group had historical clinical signs associated with EPI including weight loss/cachexia, diarrhea, or both. As we were not able to conduct physical examinations to obtain body/muscle condition scores for the EPI cats, we only have historical data related to clinical signs as reported by the primary care veterinarians and owners.

4. The authors may consider to extend the study and involve more animals with EPI to increase the robustness of the data.

We agree that including a larger number of cats would improve the statistical power and increase confidence in our findings. However, this was an unfunded pilot study, and we were limited to analyzing samples from a maximum of 6 cats with EPI and 8 healthy controls. Future studies should certainly strive to include more cats. Nonetheless, if our pilot study is published, it will be the first to examine the pathophysiology of EPI in cats making it a potentially valuable contribution to the literature. 

5. The control group was from a research colony. Did they receive any treatment or other manipulation? It would be more accurate to involve control animals that were kept in a home-like setting and receiving similar diet than EPI group.

Cats in the control group were healthy cats and to our knowledge were not receiving any therapies or experimental manipulations that could have affected the results. The cats were housed by investigators conducting studies in companion animal nutrition, but they were not currently in use for any studies utilizing dietary or medical interventions. We agree that an ideal environment for these cats would have been a home-like setting rather than a vivarium. However, we did not have access to healthy cats raised in a home environment for this study. One alternative would have been to prospectively enroll pet cats as healthy controls, however this was an unfunded pilot study and we did not have the resources to do so. 

Minor comments:

In table 1. Population statistics the unit of the age (year or month) shall be highlighted. Also, sex and weight shall be presented in this table.

We have updated the table as requested and added additional population data, but we do not have weight data on all cats, so this variable was omitted. The table contains descriptive statistics for numerical variables, whereas sex/gender is a categorical variable for which it is not possible to calculate measures of central tendency and deviation. We have represented the sex of the cats in the paragraph that precedes the table. 

Reviewer #2: The article describes the identified significant differences in serum metabolites and metabolic pathways in cats with exocrine pancreatic insufficienty (EPI) and healthy controls. I enjoyed reading it, as it is well explained and easy to follow. I really liked that the authors have the data available in Mendeley and the R code used to analyze it is available in github. However, some steps of the statistical analysis need to be revised prior publication. I recommend a major revison.

Reviewer #2, thank you for your thoughtful comments and compliments with respect to the manuscript. The authors believe that full transparency is important for any investigation involving high-throughput data and analyses involving customized statistical programming. We hope that our responses to your comments and our revisions address your concerns about the manuscript. 

Specifically:

Untargeted analysis of serum metabolomes. Authors explain that it was done by UPLC with a commercial laboratory. Authors should state which column was used for the analysis. And which extraction method. Is not the same a methal extraction than a biphasic one. And is not the same using a hilic or a csh column.

We have updated the methods section to include additional details about the UPLC. 

The selection of method for handling missing values can significantly affect subsequent data analyses and interpretations. Why the authors have used the minimum? In metabolomics, usually is used the half-minimum or kNN, pqn, ...

This is indeed an important topic. The data was generated and processed by Metabolon Inc. using a proprietary bioinformatics pipeline. I asked the data scientists at Metabolon why the minimum-value imputation method was chosen over others. Based on their experience using simulated data and real data for over 10,000 studies, Metabolon selects methods of normalization and missing values that are optimized for each study. 

An unpublished, internal studies examining different imputation methods was conducted by Metabolon. The results of these studies are now included as a supplementary file (S1File). Based on their validation studies, minimum-value imputation has the best performance for data generated on their global UPLC-MS/MS platform. 

Figure 1B fEPI group only show 4 red dots corresponding to 4 samples when there are 5 cases.

Two cats (fEPI1 and fEPI2) had very similar PCA loadings making it difficult to distinguish the two samples on the ordination plot. All 5 samples are present in the plot, but the size of the points was formatted such that they partially overlapped. We have revised Fig 1B to make it easier to distinguish the two samples by adding text annotation to each point on the PCA plot.

Did the authors conduct an exploratory PCA wihtout any normalization or scaling to check for possible outliers?

We conducted PCA without scaling and have attached an image of the ordination plot with ellipses that represent the 95% confidence interval of the multivariate T-distribution (uploaded as “Unscaled PCA for Outlier Detection”). There is still clear separation of the EPI and healthy control groups along PC1 which accounts for >36% of the total variance. Additionally, there are no samples outside of the 95% CI ellipse for each group. As such, this analysis of PCA without scaling supports the phenotypic distinction between the EPI and healthy control groups and is not consistent with the presence of outliers. There is clearly more variability in serum metabolomes among the EPI group than healthy controls and we discuss reasons for this observation in our “Discussion” section. We cannot exclude the possibility that metabolic alterations associated with EPI drive increased variability in metabolomes among the EPI cats. Given the small number of samples, it is probably not possible to perform a robust outlier detection, especially considering the greater variability among the EPI cats. As this was a pilot study, we believe that inclusion of all EPI samples is justified and important to generate new data on this poorly studied disorder. 

False discovery rate FDR<0.2 is rather big. This means 20% of false discovery rate is allowed. Why the authors didn’t use a lower FDR?

We selected a relatively liberal FDR threshold of 0.2 because the primary purpose of this pilot study was to generate novel hypotheses related to EPI in cats. We sought to strike a balance between avoiding false discoveries and missing the discovery of actual differences that could inform future targeted investigations. While many investigators use a more conservative FDR threshold of <0.05, there are numerous examples of untargeted metabolomics studies that have used a more liberal FDR threshold between 0.1 and 0.2 that have been published in PLoS ONE [PLoS ONE 11(3): e0152126, PLoS ONE 14(2): e0211762, PLoS ONE 8(10): e77629, PLoS ONE 12(5): e0177513, PLoSONE 13(6):e0197910, PLoS ONE 15(8): e0237579]. There are two general approaches to utilizing the FDR: 1) generating FDR-adjusted p-values and 2) to inform the interpretation of p-values by providing an estimate of false positives for a given level of significance. For our pilot study we chose to apply the second approach using the “qvalue” R package to calculate the FDR. However, we also recognize the importance of not reporting findings that are likely to be false positives. Thus, we have chosen to exclude results with FDR>0.2 and have further limited our “significant” findings to metabolites with a relatively large effect size to reduce the likelihood of generating spuriously “significant” results with a small effect size. Prompted by your question, we examined the distribution of FDR values in the test for significantly altered serum metabolites. Out of 197 significantly altered metabolites, 71% (140) had FDR values <0.05 and the maximum FDR was 0.0867. Had we used a more conservative FRD threshold, most of our significant results would have remained. If we accept that 8.67% of all 197 “significant” results are false positives, there would be only 18 metabolites that were falsely identified as significant in the present study. For the correlation data with fTLI 100% of FDR values were <0.05. Based on these results, we are confident that our decisions with respect to “significance” thresholds for the FDR and effect size are reasonable for a pilot study and that given the large number of significantly altered metabolites, our approach has limited the likelihood of false discovery to a reasonable degree. 

Why if the control group was constructed from a research cats colony, all the selected ones are males? In the EPI group there are also females. Gender is not important to the authors? Specific gender differences are seen in metabolomics, and this can induce a bias in the study.

We acknowledge the potential for sex and reproductive status to affect the results and we discussed this in detail (including citing relevant literature) in our “Discussion” section. This was an unfunded pilot study using a combination of prospectively collected samples (EPI group) and residual sample material from an unrelated investigation (healthy controls). For the healthy cats we only had access to residual serum from eight male cats that were collected for an unrelated metabolomics study. In our “Discussion” section, we caution the reader that our results could have been affected by differences in sex between the groups, especially with respect to the amino acids. Had this been a funded study with financial resources that allowed prospective enrollment of healthy cats and more rigorous clinical screening, we would have matched EPI cases and controls based on demographic and clinical characteristics. Unfortunately, this was not possible given the limited resources available for the present study. 

Table 1: please use the same significant digits. If they are calculated with 2 signigicant digits (e.x. mean 39.53) give 2 significant digits for the SD (ex. 18.306 should be 18.31). Be consistent in all the talbe fo the same variable (all age results with 1 significant digit, all fTLI with 2 significant digits).

We have updated the table as requested. 

Figures are pixelated. Should be uploaded with better quality.

We have uploaded only high-resolution figures (>600 dpi) and used the PACE tool to format and compress the images, as instructed in the author guidelines. The authors cannot explain the pixelated images. It is possible the images you are viewing were further compressed and pixelated during the assembly of the PDF for review. Has the reviewer attempted to open the individual source files for each figure? It seems that when the figures are viewed in the compiled pdf of the manuscript, they are pixelated, but appear in high resolution when individual files are opened and viewed. We are happy to alter the images, if requested by the Editor. 

Table 2. Some compounds are repeates with a [1] or [2] at the end...what do this numbers [1] and [2] mean?

Repeated metabolites listed in the results are due to the presence of compounds with two isomeric forms where the two isomeric species cannot be distinguished using the global UPLC platform. The numbers “[1]” and “[2]” are arbitrary designations for the two isomeric species of the same molecule that were measured. We have added captions to the tables clarifying the meaning of the repeated metabolite measurements and why the brackets are present. 

Table 2 and 3 are rather long to be in the main manuscript. I recommend to include them in the supplementary material.

These tables have been removed from the manuscript and added to the supplemental material as S4 File.

---

## [Decision Letter · Decision Letter 1]

13 Sep 2021

Untargeted analysis of the serum metabolome in cats with exocrine pancreatic insufficiency

PONE-D-21-17436R1

Dear Dr. Barko,

We’re pleased to inform you that your manuscript has been judged scientifically suitable for publication and will be formally accepted for publication once it meets all outstanding technical requirements.

Kind regards,

Francisco X. Real

Academic Editor

PLOS ONE

Additional Editor Comments (optional):

Reviewers' comments:

Reviewer's Responses to Questions

**Comments to the Author**

1. If the authors have adequately addressed your comments raised in a previous round of review and you feel that this manuscript is now acceptable for publication, you may indicate that here to bypass the “Comments to the Author” section, enter your conflict of interest statement in the “Confidential to Editor” section, and submit your "Accept" recommendation.

Reviewer #1: All comments have been addressed

Reviewer #2: All comments have been addressed

2. Is the manuscript technically sound, and do the data support the conclusions?

Reviewer #1: Yes

Reviewer #2: Yes

3. Has the statistical analysis been performed appropriately and rigorously? 

Reviewer #1: Yes

Reviewer #2: Yes

4. Have the authors made all data underlying the findings in their manuscript fully available?

Reviewer #1: Yes

Reviewer #2: Yes

5. Is the manuscript presented in an intelligible fashion and written in standard English?

Reviewer #1: Yes

Reviewer #2: Yes

6. Review Comments to the Author

Reviewer #1: The authors replyed to my comments and the manuscript imporved. Some major limitations remianed that are listed in the discussion.

Reviewer #2: (No Response)

7. PLOS authors have the option to publish the peer review history of their article (what does this mean?). If published, this will include your full peer review and any attached files.

Reviewer #1: **Yes: **József Maléth

Reviewer #2: **Yes: **Raquel Cumeras

---

## [Editor Report · Acceptance letter]

22 Sep 2021

PONE-D-21-17436R1 

Untargeted Analysis of the Serum Metabolome in Cats with Exocrine Pancreatic Insufficiency 

Dear Dr. Barko:

I'm pleased to inform you that your manuscript has been deemed suitable for publication in PLOS ONE. Congratulations! Your manuscript is now with our production department. 

Kind regards, 

on behalf of

Dr. Francisco X. Real 

Academic Editor

PLOS ONE